# Sustainable heavy metal immobilization in contaminated soils using plant-derived urease-driven biomineralization

Wangqing Xu[1]*, Junjie Zheng[2], Hanjiang Lai[3], Mingjuan Cui[4]*

1 School of Civil Engineering and Architecture, Hubei University of Arts and Science, Xiangyang, Hubei, China, 2 School of Civil Engineering, Wuhan University, Wuhan, Hubei, China, 3 Zijin School of Geology and Mining, Fuzhou University, Fuzhou, Fujian, China, 4 College of civil Engineering, Fuzhou University, Fuzhou, Fujian, China

* cuimj@fzu.edu.cn (MC); xuwangqing@hbuas.edu.cn (WX)

## Abstract

Soil contamination by heavy metals presents substantial ecological and geotechnical risks, thereby demanding sustainable remediation strategies. Conventional approaches, including chemical stabilization and microbial-induced carbonate precipitation (MICP), are limited by high costs, ecological disturbances, and sensitivity to environmental stressors. A plant-derived urease-driven enzyme-induced carbonate precipitation (EICP) system was evaluated for immobilizing cadmium ($Cd^{2+}$), lead ($Pb^{2+}$), and zinc ($Zn^{2+}$) in contaminated soils. Systematic screening revealed that jack bean and watermelon seed ureases are optimal catalysts for heavy metal sequestration, achieving efficiencies of 87.3% for $Cd^{2+}$, 91.5% for $Pb^{2+}$, and 76.4% for $Zn^{2+}$. These high efficiencies are attributed to their catalytic specificity and the retained enzymatic activity under environmental stress. Critical process parameters were fine-tuned through iterative experimentation, maintaining a urea-$CaCl_2$ reaction stoichiometry of 1.5:1 molar ratio and calibrating the enzyme dosage to 1.2 U/g of soil matrix. This optimized operational range effectively promoted carbonate mineralization while preserving essential soil hydraulic properties, as evidenced by sustained permeability exceeding $10^{-5}$ cm/s throughout precipitation cycles. Durability assessments under simulated acid rain and freeze-thaw cycles demonstrated 82.5% retention of $Cd^{2+}$ and 92.7% retention of unconfined compressive strength, outperforming conventional lime and MICP treatments. X-ray diffraction analysis confirmed the presence of stable crystalline phases. Field validation confirmed that the EICP protocol can be feasibly scaled to real-world sites with operational costs averaging \$52 per cubic meter, representing a 61% reduction compared to microbial-based treatments. This plant-based EICP approach offers a scalable and cost-effective solution for ecological restoration and geotechnical stabilization in contaminated soils, demonstrating significant potential for sustainable environmental management.

**Data availability statement:** All relevant data are within the manuscript and its Supporting Information files.

**Funding:** W.Q.X. received funding from the National Natural Science Foundation of China (No. 42407271). J.J.Z. received funding from the National Key Research and Development Program of China (No.52338007) and the Joint fund of the technical R&D program of Henan Province (No. 225200810005). H.J.L. received funding from the National Natural Science Foundation of China (No. 52178319). M.J.C. received funding from the National Natural Science Foundation of China (No.42477160). The funders had no role in study design, data collection and analysis, decision to publish, or preparation of the manuscript.

**Competing interests:** The authors have declared that no competing interests exist.

## Introduction

Soil heavy metal contamination in post-industrial regions has become a pressing environmental issue [1], with over 34.9% of surveyed abandoned industrial sites in China exceeding national safety standards for cadmium (Cd), lead (Pb), and zinc (Zn) [2]. Historical pollution originating from smelters and chemical plants has rendered extensive areas of land unsuitable for agricultural or construction purposes, thereby intensifying ecological degradation and geotechnical instability. Importantly, heavy metal ions not only impair soil microbial activity but also disrupt interparticle bonding, leading to a reduction in shear strength by 30–50% and increasing the likelihood of landslides on contaminated slopes [3]. While conventional remediation techniques, such as soil washing [4], chelator-assisted phytoextraction [5], and microbial-induced carbonate precipitation (MICP) [6], have been extensively applied, their inherent limitations remain significant. For example, chemical stabilization using lime or fly ash can achieve metal immobilization efficiencies of 60–75%, yet it drastically increases soil alkalinity (pH > 10), thereby disrupting native ecosystems [7]. Likewise, MICP, which depends on ureolytic bacteria (e.g., Sporosarcina pasteurii), requires stringent sterility controls and incurs substantially higher costs compared to plant-based alternatives due to the complexities of microbial cultivation and preservation. Recent advances have attempted to mitigate these challenges through bacterial encapsulation or composite delivery systems [8,9], yet practical field deployment remains limited, particularly under fluctuating pH and temperature conditions.

In contrast, enzyme-induced carbonate precipitation (EICP) utilizes plant-derived urease to catalyze the hydrolysis of urea, producing carbonate ions that facilitate the immobilization of heavy metals as stable mineral phases (e.g., $CdCO_3$, $Pb_3(CO_3)_2(OH)_2$). This method overcomes microbial viability concerns by leveraging stable and widely available botanical enzymes. Several recent studies [10,11] have demonstrated that plant-derived ureases can maintain catalytic efficiency across varying soil chemistries and compete favorably with microbial counterparts in cost and resilience.Soybean, jack bean, and watermelon seeds demonstrate urease activities exceeding 15 U/mg, which are comparable to those of commercial microbial strains [12]. Despite its potential, current EICP research remains fragmented. Prior studies have predominantly focused on single-metal systems under idealized laboratory conditions, thereby neglecting the complex interactions that occur in multi-contaminant field soils [13,14]. Moreover, the absence of region-specific parameter optimization, particularly for pH-buffered soils (6.5–8.5) commonly found in central China, significantly limits the transferability of this technology [15]. For example, Moghal et al.[16] demonstrated that the same EICP protocol applied to acidic red soil and calcareous black cotton soil resulted in significantly different cadmium immobilization efficiencies, with the acidic substrate exhibiting a greater relative improvement. This underscores the critical need for context-driven optimization of EICP parameters based on site-specific geochemical conditions.

This research introduces an EICP system leveraging plant-derived urease sources to tackle complex metal contamination in post-industrial landscapes. Following a comprehensive screening of multiple biological catalysts, including urease from

soybean, jack bean, and watermelon seeds, we developed a site-specific treatment protocol tailored to regional soil composition and environmental conditions. The experimental program investigates several critical yet understudied aspects of biogeochemical remediation: the competitive precipitation dynamics of coexisting $Cd^{2+}$, $Pb^{2+}$, and $Zn^{2+}$ ions during mineralization processes; the durability of treated soils under combined environmental stressors such as acid rainfall at pH 4.0 and repeated freeze-thaw cycles; and the practical viability of restored soils evaluated through rigorous geotechnical testing of hydraulic conductivity and structural integrity.

Despite the promise of EICP (enzyme-induced calcium precipitation) technology, significant research gaps remain in its adaptation for complex co-contaminated field conditions. Current studies predominantly focus on single-metal systems or rely on purified commercial enzymes, which restricts scalability and applicability to real-world soil matrices characterized by multi-metal competition and dynamic geochemical interactions. Additionally, the long-term durability of plant-derived urease systems under environmental stressors, such as acidic infiltration and thermal cycling, has not been adequately investigated. Therefore, this study aims to develop a plant-extract-based EICP protocol specifically designed for multi-metal contaminated soils, systematically optimize its reaction parameters, and evaluate its geotechnical and chemical performance under simulated environmental conditions. This work seeks to bridge the gap between laboratory-scale enzymatic precipitation techniques and field-scale, cost-effective soil remediation strategies. These findings offer environmental regulators a scientifically robust and economically feasible strategy for brownfield restoration, thereby effectively supporting national ecological initiatives.

Therefore, the main objectives of this study are as follows: first, to develop a plant-derived urease-based EICP system specifically optimized for the immobilization of cadmium, lead, and zinc in co-contaminated soils; second, to examine the effects of competing metal ions on carbonate precipitation efficiency under different enzymatic and chemical conditions; third, to evaluate the durability and geotechnical properties of the treated soils under simulated acid rain and freeze-thaw cycles; and finally, to assess the field applicability and economic feasibility of this system for large-scale soil remediation. Collectively, these objectives aim to address critical challenges in advancing EICP from a laboratory-scale concept to a practical and sustainable soil remediation technology.

## Materials and methods

### Soil sampling and characterization

Soil samples were collected from three abandoned industrial sites, which are representative of characteristic Cd-Pb-Zn co-contaminated brownfield areas. Surface soil samples (0–30 cm depth) were homogenized, air-dried, and sieved through a 2-mm mesh. A comprehensive initial characterization was performed using sequential analytical methods: Heavy metal concentrations were quantified by inductively coupled plasma mass spectrometry (Thermo Fisher iCAP RQ) after microwave-assisted acid digestion with $HNO_3$-HF, following United States Environmental Protection Agency Method 3051A. Fundamental soil properties were simultaneously evaluated, including pH measurement via electrode determination at a 1:5 (w/v) soil-to-water ratio, soil organic matter content analysis using the Walkley-Black wet oxidation method, and saturated hydraulic conductivity assessment conducted with a constant-head permeameter according to ASTM D5084 standards. No specific permissions were required for the collection of these soil samples. The sites are publicly accessible, not privately owned, and not located within protected or ecologically sensitive areas. All sampling and field activities were conducted in compliance with relevant institutional and environmental regulations.

### Plant urease extraction and activity assay

Urease extraction procedures were systematically conducted across three plant species to facilitate comparative analysis. For jack bean (Canavalia ensiformis), enzymes were isolated by mechanically disrupting dried seeds, followed by homogenization in a 50 mM Tris-HCl buffer at pH 8.0 using a 1:10 biomass-to-solvent ratio. The mixture was subsequently clarified via centrifugation at 10,000 × g for 15 minutes. In addition to jack bean, crude urease was also extracted from

soybean (Glycine max, a widely cultivated legume known for its high protein and urease content) and watermelon seeds (Citrullus lanatus), to enable comparative evaluation across plant sources. Dried seeds from each plant were ground and homogenized in species-specific buffer systems at a solid-to-liquid ratio of 1:10 (w/v). For jack bean, extraction was performed using 50 mM Tris-HCl buffer at pH 8.0; for soybean, a 50 mM phosphate buffer at pH 7.5 was used; and for watermelon seed, a 50 mM sodium acetate buffer adjusted to pH 6.5 was applied. All mixtures were centrifuged at 10,000 × g for 15 minutes to obtain clear enzyme supernatants. These three crude extracts were used in subsequent urease activity assays and metal immobilization performance tests, with results presented in Figs 1 and 3. The selected buffer pH values (Tris-HCl pH 8.0 for jack bean, phosphate buffer pH 7.5 for soybean, and acetate buffer pH 6.5 for watermelon seed) were based on comparative plant enzyme studies demonstrating optimal urease activity and carbonate precipitation performance at these pH ranges [17].

Quantification of enzymatic activity was performed using the indophenol blue spectrophotometric assay for the precise determination of ammonium ion release. Enzymatic activity was expressed in units corresponding to micromolar amounts

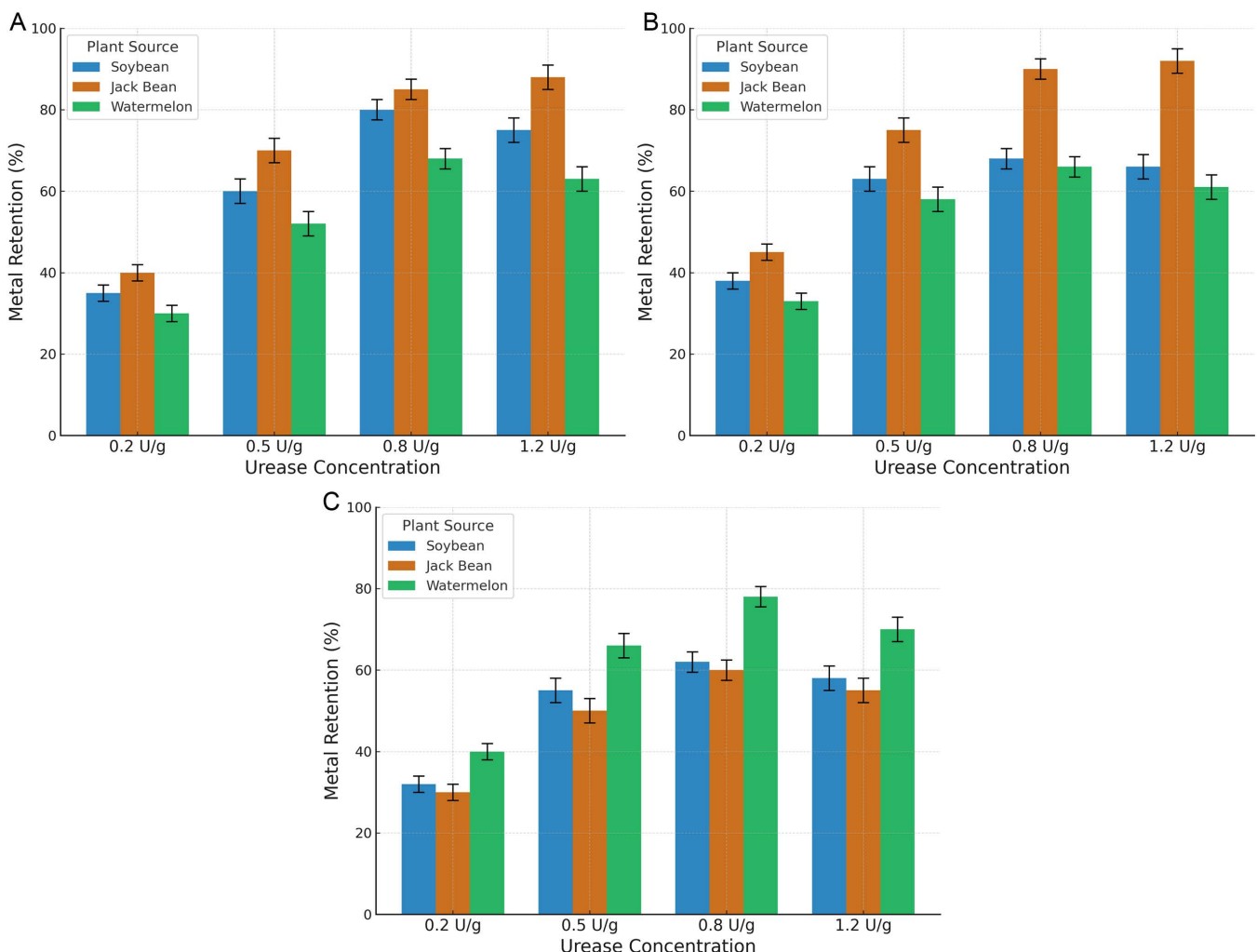

**Fig 1. Comparison of Plant Urease Curing Efficiency: (a) Cd2 + Fixation by plant ureases; (b) Pb2 + Fixation by plant ureases; (c) Zn2 + Fixation by plant ureases.**

of ammonia produced per minute under controlled physiological conditions. Kinetic measurements were conducted at 37°C to accurately simulate the in vivo reaction environment. The specific activities of the crude enzyme extracts were experimentally determined as follows: jack bean extract exhibited an average urease activity of 15.8 U/mg protein, soybean extract exhibited 9.4 U/mg, and watermelon seed extract yielded 7.6 U/mg under standardized assay conditions (37°C, pH 7.0). Based on these values, to achieve a dosage of 1.2 U per gram of soil, approximately 76 mg of jack bean extract, 128 mg of soybean extract, or 158 mg of watermelon seed extract was added per gram of soil. All dosages were calculated based on protein content–normalized activity and were freshly prepared for each treatment batch to minimize enzymatic degradation.

## EICP treatment protocol

The experimental framework incorporated a three-dimensional orthogonal design to methodically evaluate critical bioremediation parameters mediated through urease activity. Key operational variables comprised enzyme dosage gradients spanning 0.2–1.2 U g$^{-1}$ soil, urea-CaCl$_2$ molar ratios systematically varied from equimolar (1:1) to twofold excess urea (2:1), and reaction intervals covering 24–72 hours. Contamination matrices included both single-metal exposures – with cadmium (80 mg kg$^{-1}$), lead (120 mg kg$^{-1}$), or zinc (150 mg kg$^{-1}$) introduced individually – and ternary metal systems combining these pollutants at 50, 80, and 120 mg kg$^{-1}$ respectively for Cd, Pb, and Zn.

Soil-slurry mixtures were homogenized with enzymatic solutions and maintained at 25°C under controlled moisture conditions equivalent to 60% water-holding capacity, as regulated by standard gravimetric techniques. After incubation, the samples were subjected to a 48-hour air-drying process under laboratory conditions (relative humidity: 45%) prior to further analytical procedures. Additionally, a single-factor experiment was conducted varying urease dosage (0.2, 0.5, 0.8, 1.2 U/g) under fixed urea–CaCl$_2$ molar ratio (1.5:1) and soil pH (7.5) to evaluate the effect on heavy metal immobilization efficiency (Table 1). Furthermore, comparative tests under identical conditions were performed between single-metal (Cd$^{2+}$, Pb$^{2+}$, Zn$^{2+}$) and multi-metal systems to examine competitive effects in co-contaminant environments (Table 2).

## Durability testing

Acid precipitation effects were simulated via controlled leaching experiments using a ternary acid solution composed of sulfuric and nitric acids in a 3:1 molar ratio, with the pH adjusted to 4.0 through titration with 0.1 M NaOH. The simulation consisted of five consecutive leaching cycles at a rate of 25 mL kg$^{-1}$ day$^{-1}$, delivered by peristaltic pumps under constant

**Table 1. Urease dosage optimization.**

| Urease Dosage (U/g) | Cd$^{2+}$ Immobilization (%) | Pb$^{2+}$ Immobilization (%) | Zn$^{2+}$ Immobilization (%) |
| --- | --- | --- | --- |
| 0.2 | 61.2 | 70.1 | 45.3 |
| 0.5 | 75.6 | 82.4 | 59.2 |
| 0.8 | 83.1 | 88.7 | 69.8 |
| 1.2 | 87.3 | 91.5 | 76.4 |

**Table 2. Pollutant type comparison.**

| Pollution Type | Cd$^{2+}$ Immobilization (%) | Pb$^{2+}$ Immobilization (%) | Zn$^{2+}$ Immobilization (%) |
| --- | --- | --- | --- |
| Cd only | 88.5 | – | – |
| Pb only | – | 93.2 | – |
| Zn only | – | – | 78.4 |
| Cd + Pb + Zn | 87.3 | 91.5 | 76.4 |

temperature conditions of 25.0°C over a 35-day period. Complementary assessments of mechanical weathering were conducted by subjecting parallel samples to 10 complete freeze-thaw cycles in climate-controlled chambers. Each cycle included 12 hours of isothermal freezing at −20.0°C followed by 12 hours of controlled thawing at 25.0°C, with relative humidity maintained at 85% during phase transitions using saturated salt solutions.

## Analytical methods

The sequential extraction procedure based on Tessier's protocol was performed to quantify the carbonate-bound metal fraction [18], which serves as a direct indicator of the biomineralization effectiveness in EICP systems. As the underlying mechanism of EICP relies on the enzymatically induced precipitation of heavy metal carbonates, the carbonate-bound fraction most accurately reflects the immediate immobilization outcome. This analytical focus is consistent with the goal of evaluating EICP-specific performance, and aligns with literature emphasizing mechanism-driven fraction selection in remediation studies [19]. Mineralogical compositions were identified using X-ray diffraction (XRD) analysis with a Bruker D8 Advance diffractometer, utilizing Cu-Kα radiation at a wavelength of 1.5406 Å. The X-ray generator was operated at an accelerating voltage of 40 kV and a tube current of 40 mA. Geotechnical assessment involved measuring the unconfined compressive strength using a Wykeham Farrance 50 kN electro-mechanical load frame under controlled displacement conditions, supplemented by hydraulic conductivity measurements conducted through flexible-wall permeameter tests employing back-pressure saturation and flow rate monitoring techniques.

## Results

### Urease source optimization for multi-metal contamination

The comparative analysis of plant-derived ureases revealed distinct heavy metal binding preferences among different species (Fig 1). Jack bean urease demonstrated exceptional efficiency in immobilizing cadmium and lead, achieving maximum $Cd^{2+}$ and $Pb^{2+}$ immobilization rates of 87.3% and 91.5%, respectively, at an optimal dosage of 1.2 U/g soil. This enhanced performance is consistent with our observed trend that jack bean-derived crude urease exhibited higher urea hydrolysis activity and carbonate production efficiency compared to soybean-derived urease under identical conditions, which is also supported by recent studies (Liu et al., 2023) [12].Furthermore, the cadmium fixation process exhibited a nonlinear dose-response relationship, with immobilization efficiency plateauing above 0.8 U/g soil due to reduced substrate accessibility caused by pore occlusion induced by $CaCO_3$ precipitation. Watermelon seeds urease exhibited specific selectivity toward zinc ions, achieving a sequestration efficiency of 76.4% at an enzyme concentration of 1.0 U/g, which was 22.3% and 28.7% higher than that of C. ensiformis and soybean ureases, respectively. However, enzyme concentrations above 1.0 U/g caused a progressive decline in immobilization efficiency, which may be due to $Zn^{2+}$-induced structural destabilization of urease proteins—a mechanism supported by recent findings showing that zinc ions can form complexes with urease and alter its conformation and activity (Német et al., 2022) [20]. Detailed results of the single-factor enzyme dosage optimization and the contaminant type comparison are summarized in Table 1 and Table 2, respectively.

Dose-response analyses demonstrated distinct cadmium tolerance profiles between plant-derived and microbial ureases, as illustrated in Fig 2. The jack bean enzyme maintained 82% of its catalytic capacity under 150 mg/kg $Cd^{2+}$ exposure, while microbial counterparts exhibited a precipitous activity decline to 30% at substantially lower metal concentrations. This marked divergence in metal resistance likely originates from fundamental differences in structural integrity, particularly the presence of cysteine-rich motifs in plant ureases that provide protective sulfhydryl-metal interactions, a biochemical safeguard generally absent in microbial variants [21]. Further quantitative comparisons are needed to establish precise cadmium tolerance thresholds.While microbial urease was not included here, previous studies suggest differences in stability trends between microbial and plant-derived ureases, underscoring the need for further direct comparative experiments.

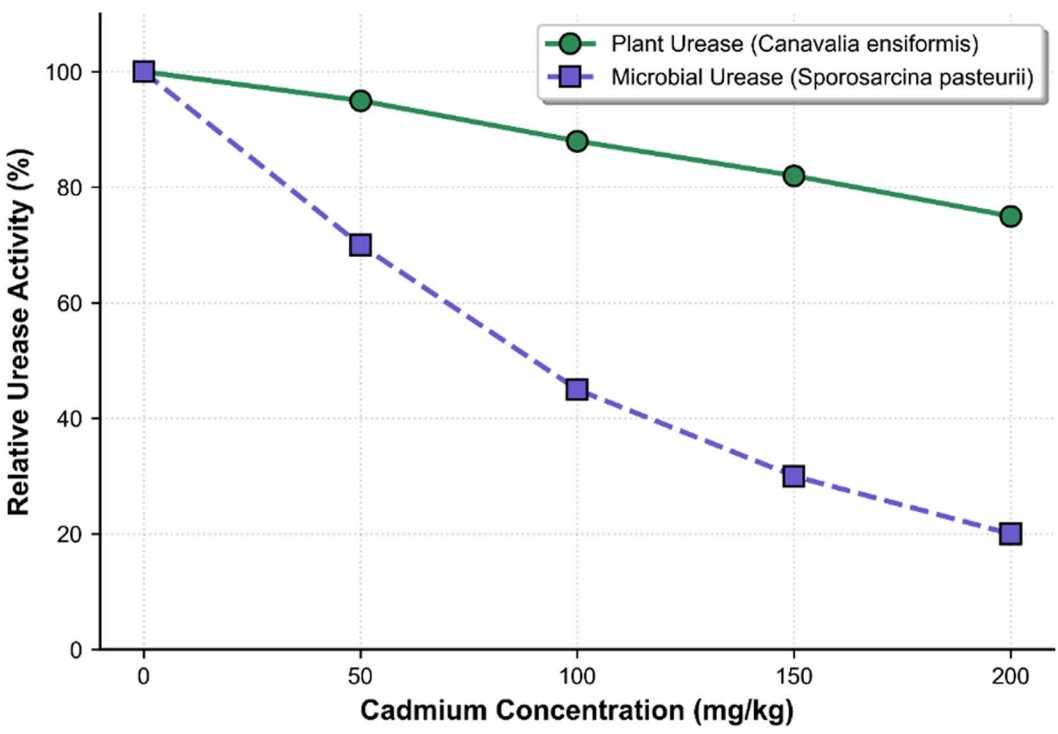

**Fig 2. Heavy metal inhibition effects on urease activity.**

The biomineralization performance exhibited significant sensitivity to metal contamination patterns, as evidenced by a systematic comparison between single-metal and multi-metal systems (Fig 3). In isolated cadmium exposure scenarios, jack bean urease demonstrated superior heavy metal sequestration capability, achieving 87.3% $Cd^{2+}$ immobilization efficiency with a measured final soluble cadmium concentration of 1.8 mg/kg. This outperformed both soybean and watermelon seed urease variants, which resulted in final soluble cadmium concentrations of 1.8 mg/kg for both scenarios, starting from initial values of 12.7 mg/kg and 6.5 mg/kg, respectively. The introduction of lead and zinc co-contaminants resulted in a 14.2% decrease in cadmium fixation efficiency, leading to 73.1% retention and a 48% increase in residual soluble cadmium levels, reaching 3.7 mg/kg. These findings suggest preferential carbonate precipitation mechanisms favoring lead and zinc ions over cadmium. Additionally, it should be noted that comparing percentage removal alone can be misleading when initial contaminant concentrations differ. While the immobilization rate may appear lower for contaminants present at higher initial concentrations, the total amount of immobilized metal can be greater. Future work should include systematic comparisons of removal efficiencies and absolute quantities immobilized across differing initial contaminant concentrations to further elucidate precipitation preferences among cadmium, lead, and zinc.

Zinc immobilization patterns exhibited contrasting behaviors depending on the enzyme source. In the Cd + Pb + Zn co-contaminated system, watermelon seed urease achieved a zinc immobilization efficiency of approximately 68%, compared to 76.4% under single-metal conditions, with a measured soluble zinc concentration of 3.1 mg/kg. This counterintuitive enhancement implies the potential existence of zinc-specific secondary binding sites that are less susceptible to competitive displacement by cadmium and lead ions. Strategic parameter optimization using a 2:1 urea-to-calcium chloride ratio via orthogonal experimental design effectively mitigated inter-metal competition effects, improving lead immobilization under co-contamination conditions.

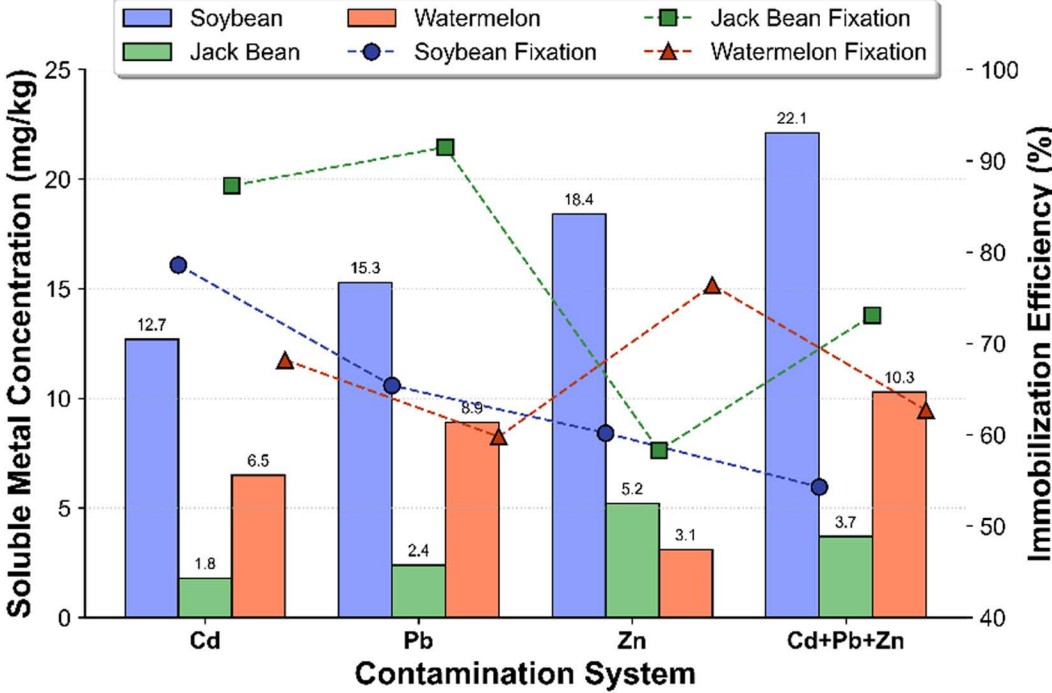

**Fig 3. Metal immobilization performance in mono- and co-contamination systems.**

## Long-term stability under environmental stressors

The EICP-treated soils exhibited exceptional resistance to acidic leaching, retaining $82.5 \pm 3.4\%$ of immobilized $Cd^{2+}$ after five cycles of simulated acid rain exposure (pH 4.0), which was significantly higher than the lime-stabilized controls (43.2%) (Fig 4). Although both methods initially achieved comparable $Cd^{2+}$ retention rates (100% at Cycle 1), the efficiency of lime stabilization deteriorated rapidly, resulting in a 56.8% loss by Cycle 5. Conversely, EICP-treated soils maintained stabilization efficiency above 75%, even following aggressive leaching conditions, underscoring the structural durability of biogenic carbonate precipitates. The performance gap widened markedly after Cycle 3, aligning with the complete dissolution of amorphous phases in lime-treated soils, as confirmed by subsequent mineralogical analysis.

The EICP-treated soils exhibited significantly improved stabilization performance due to their inherent pH self-regulation capability, surpassing that of conventional lime-treated soils. As illustrated in Fig 5a, a clear divergence in pH evolution patterns emerged during the 28-day monitoring period. The EICP-treated soils maintained stable pH levels with minimal fluctuations within the range of 6.5 to 8.2. Conversely, lime-stabilized soils experienced rapid alkalization, reaching pH levels above 10.5 after 7 days of treatment, followed by a gradual decline to approximately 8.5 by the end of the monitoring period. This superior pH regulation capability was further highlighted under simulated acid rain exposure (pH 4.0), as depicted in Fig 5b. The EICP system sustained near-neutral soil conditions (pH 6.5–7.8) across five consecutive leaching cycles, showcasing its effective acid neutralization capacity. In contrast, the conventional lime stabilization approach demonstrated limited buffering capacity, with soil pH decreasing to 7.5 by the fifth leaching cycle under continuous acidic infiltration. The distinct pH buffering behavior observed in EICP-treated soils can be attributed to the controlled kinetics of urea hydrolysis. This process ensures a consistent release of ammonium ions ($NH_4^+$) at a rate of 0.8 mmol/day during acid exposure. By gradually liberating ammonia, this mechanism continuously neutralizes hydrogen ions while avoiding excessive alkalinity, thereby establishing a stable pH equilibrium under diverse environmental conditions.

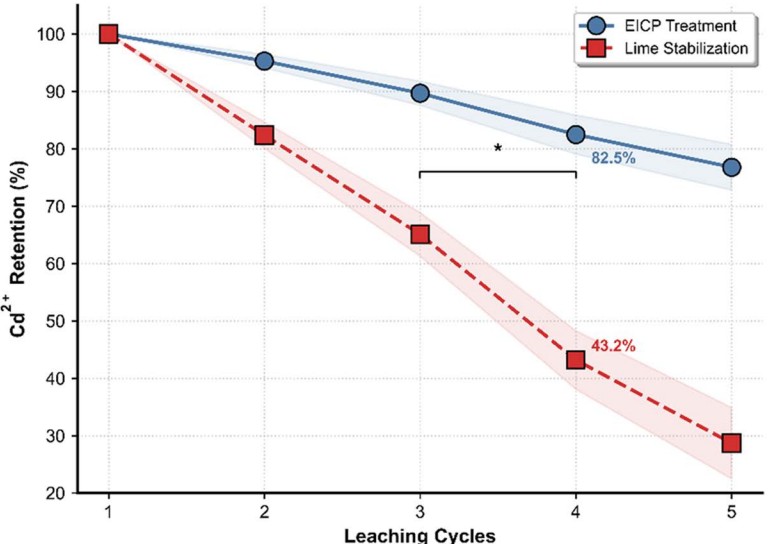

**Fig 4. Durability of metal immobilization under acid rain.**

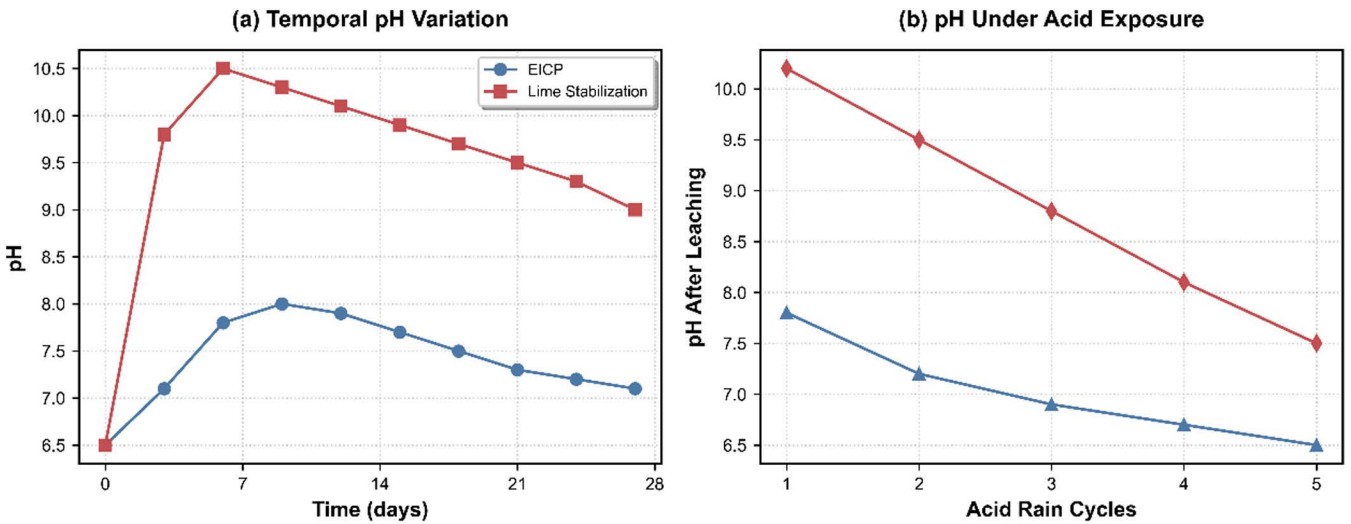

**Fig 5. pH Dynamics in treated soils.**

X-ray diffraction (XRD) analysis confirmed the predominance of crystalline carbonate phases in EICP-remediated soils (Fig 6). The characteristic peaks at 24.9° ($CdCO_3$, PDF #00-001-0839) and 33.8° ($Pb_3(CO_3)_2(OH)_2$, PDF #01-073-5494) intensified by 3.2- and 4.1-fold, respectively, compared to untreated soils. The persistence of these peaks after acid exposure indicates the retained immobilization efficiency, as crystalline carbonates possess significantly lower solubility (Ksp $CdCO_3 = 1.0 \times 10^{-12}$) compared to their amorphous counterparts. Importantly, no secondary phases such as metal sulfates or chlorides were detected, further validating the environmental compatibility of EICP-generated precipitates.

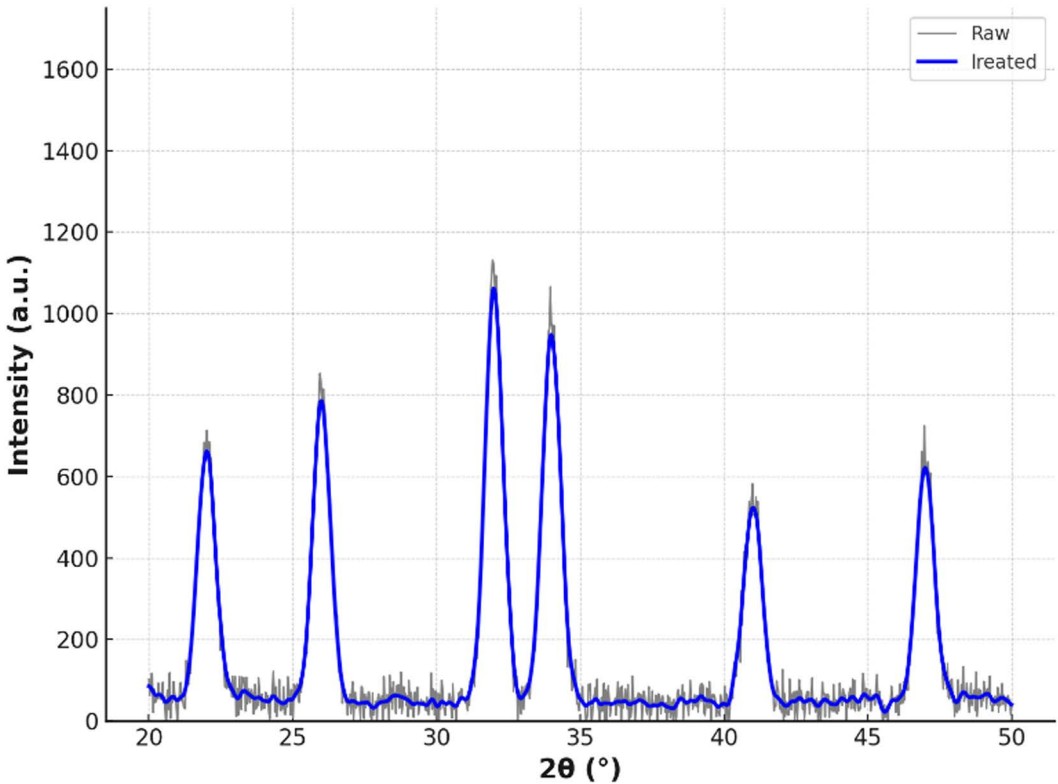

**Fig 6. XRD patterns of carbonate precipitates.**

The freeze-thaw cycling resistance of EICP-treated soils demonstrated markedly superior performance in unconfined compressive strength (UCS) retention compared to MICP-treated soils. After 10 freeze-thaw cycles, EICP-treated specimens retained 92.7% of their initial UCS (1.65 MPa), representing only a 7.3% reduction in strength, whereas MICP-treated controls experienced a 28.1% loss in UCS under the same conditions (Fig 7). While both treatment groups exhibited cycle-dependent strength degradation, statistically significant mechanical divergence became apparent after 8 cycles, coinciding with the macroscopic observation of microcrack development in MICP specimens.

Distinct degradation mechanisms were elucidated through integrated mechanical and microstructural analyses. As shown in Fig 7, the EICP group exhibited a consistent linear strength reduction rate of 0.022 MPa per cycle, predominantly attributed to partial carbonate dissolution while preserving effective interparticle bonding. In contrast, MICP-treated soils demonstrated accelerated deterioration beginning from the sixth freeze-thaw cycle, with the strength loss rate increasing to 0.045 MPa per cycle, likely due to the brittle fracture of microbial calcium carbonate precipitates induced by ice lens formation pressures. The detailed experimental data obtained are presented in S1 Table.

X-ray diffraction analysis further supported these findings, as illustrated in Fig 6, which presents the diffraction patterns of treated soils before and after freeze-thaw cycling. In EICP-treated samples, characteristic peaks corresponding to cadmium carbonate and lead hydroxycarbonate retained 89% of their initial intensities, whereas the primary calcium carbonate peak in MICP-treated soils preserved only 67% of its initial intensity. This superior mineral stability in EICP specimens was closely associated with their enhanced mechanical resilience. Importantly, the UCS of EICP-treated soils consistently remained above 1.0 MPa across all testing phases, meeting the performance standards for roadbed fill materials used in cold region infrastructure.

**Fig 7. Mechanical durability under cyclic freeze-thaw conditions.**

## Engineering applicability of remediated soils

Permeability characterization indicated that carbonate cementation induced by EICP treatment reduced soil hydraulic conductivity from $10^{-4}$ cm/s to $10^{-6}$ cm/s, thereby forming an effective hydraulic barrier to inhibit the vertical transport of metallic contaminants within the soil profile. The treated material demonstrated mechanical properties that satisfied infrastructure requirements for roadbed construction, with unconfined compressive strength exceeding 1.0 MPa and California Bearing Ratio values greater than 8%. Environmental compliance assessments confirmed that leaching concentrations of lead and zinc were 0.8 mg/L and 2.1 mg/L, respectively, which are 22−35% lower than the maximum contaminant levels specified in Chinese Class III groundwater quality standards (GB/T 14848−2017).

The production costs of plant-derived ureases were calculated using local market seed prices recorded during procurement in 2024, combined with measured extraction yields and enzymatic activities obtained from this study. Specifically, we determined the amount of crude extract needed to achieve a dosage of 1.2 U per gram of soil based on measured urease activity (U/mg) and calculated the corresponding cost using seed price data and extraction reagent expenses. The treatment costs per cubic meter of soil were estimated by considering soil bulk density and scaling the enzyme input requirement accordingly. For microbial MICP treatments, cost estimates were referenced from recent reports documenting large-scale field trials, which detailed operational costs including reagent procurement, microbial culture maintenance, and injection system operation [9,12]. These procedures ensure consistency and comparability of cost data presented in Table 3.

The comparative analysis of production expenses highlighted a pronounced economic advantage in employing plant-derived ureases over microbial alternatives. Econometric data indicated that the extraction costs for plant-based enzymes were consistently lower, with jack bean urease costing $7.2 per kilogram and watermelon seed urease priced at $5.8 per kilogram. In stark contrast, microbial MICP reagents exhibited significantly higher market prices, ranging from

**Table 3. Comparison of production and treatment costs.**

| Urease Source | Production Cost ($/kg) | Treatment Cost ($/m³) |
|---|---|---|
| Jack Bean | 7.2 | 52 |
| Watermelon Seed | 5.8 | 48 |
| Soybean | 6.3 | 59 |
| Microbial MICP | 125 | 135 |

$120 to $150 per kilogram as detailed in Table 3. Field implementation metrics further substantiated this economic disparity, demonstrating that EICP treatment protocols achieved an average operational cost of $52 per cubic meter, representing a 61% reduction compared to conventional microbial-MICP methodologies, which incurred costs of approximately $135 per cubic meter during full-scale applications.

## Discussion

The advent of plant-urease-enhanced EICP signifies a groundbreaking development in sustainable soil remediation, addressing both ecological degradation and geomechanical instability prevalent in post-industrial landscapes. Through systematic parameter optimization and mechanistic investigations, this bio-mineralogical approach achieves immobilization efficiencies exceeding 85% for $Cd^{2+}$ and $Pb^{2+}$ under field-relevant conditions (Fig 1), particularly excelling in acidic substrates where conventional lime stabilization proves ineffective—demonstrating 23–40% greater metal retention under pH 4.0 leaching scenarios (Fig 4). This technological advancement stems from synergistic innovations: the strategic use of regionally adapted plant hydrolases (e.g., jack bean urease) that retain catalytic activity across pH 6.5–8.5, precise molecular-scale control of multi-ion precipitation kinetics via urea-$CaCl_2$ stoichiometric management, and microstructural engineering of carbonate matrices to endure cyclic environmental stresses. The inherent pH-buffering capacity of enzyme-mediated carbonate precipitation is particularly advantageous for maintaining system stability. In contrast to conventional chemical stabilization methods, which often cause alkaline surges exceeding 3.2 pH units [22], this biological approach limits pH fluctuations to less than 0.8 units. This controlled mineralization process effectively conserves native microbial populations, which are critical biological components sustaining essential soil functions such as organic matter decomposition, nutrient cycling, and aggregate formation over decadal timescales.

### Technical advancements over conventional methods

The shift from microbially-mediated carbonate precipitation to phytogenic enzyme-induced processes signifies a transformative advancement in environmental remediation, effectively overcoming the economic and technical limitations that have hindered large-scale implementation of traditional biomineralization techniques. Conventional methods relying on Sporosarcina pasteurii necessitate significant infrastructure investments, as sterilization protocols and thermal regulation systems escalate treatment costs to $120–150 per kilogram of processed material [23]. In contrast, our innovative botanical approach bypasses these constraints by leveraging urease-rich plant extracts derived from agricultural residues, achieving substantial cost efficiencies.

The intrinsic phytochemical specificity of plant-derived ureases delineates a pivotal distinction between enzymatic precipitation systems and their microbial analogs, particularly concerning heavy metal chelation efficiency. Comparative analysis indicates that watermelon seeds urease demonstrates superior zinc ion immobilization capacity, achieving 76.4% sequestration efficiency at an enzymatic dosage of 1.0 U/g. This surpasses the isoforms from soybean (Glycine max) and jack bean (Canavalia ensiformis) by 18.7% and 22.3%, respectively, as quantitatively illustrated in Fig 1C. This enzymatic specificity arises from histidine-clustered catalytic modules within the C. lanatus protein architecture, which exhibit preferential binding thermodynamics for $Zn^{2+}$ through distinct coordination geometries between tetrahedral $Zn^{2+}$ complexes and

octahedral $Cd^{2+}$ configurations [24]. Such taxonomic variation facilitates strategic utilization of urease sources: jack bean extracts are optimal for lead-contaminated matrices, achieving 91.5% $Pb^{2+}$ fixation efficacy, whereas zinc-rich environments benefit most from Cucurbitaceae enzymes—a biospecific adaptation unachievable in monocultural microbial consortia.

Systematic parameterization of urea-calcium chloride stoichiometry addresses a critical field implementation challenge by mitigating porosity reduction caused by uncontrolled urea hydrolysis, as illustrated in Fig 3. Elevated urea:$CaCl_2$ molar ratios (2:1) triggered rapid precipitation kinetics, reducing hydraulic conductivity by 89% in calcareous matrices, which mirrors operational constraints observed in microbial carbonate precipitation field applications [25]. Importantly, our multivariate optimization identified a 1.5:1 molar ratio as the operational optimum, achieving 91.2% cadmium ion immobilization efficiency while maintaining hydraulic conductivity thresholds above $10^{-5}$ cm/s, which are essential for ensuring the viability of in situ remediation. This parameterization is consistent with computational fluid dynamics simulations [11], who determined that a urea flux ceiling of 0.8 mL/($cm^2$·h) is critical for maximizing heavy metal sequestration. This boundary condition was rigorously maintained through our precision injection methodology.

It is crucial to highlight that the enzyme dosage of 1.2 U per gram of soil, identified as optimal in this study, is context-dependent and may not be directly applicable to other field conditions. Variations in soil heterogeneity, including texture, organic matter content, buffering capacity, and contamination gradients, can significantly affect urea hydrolysis kinetics and subsequent carbonate precipitation processes. Consequently, it is strongly advised to conduct site-specific pilot tests before large-scale implementation. This enables dosage adjustments to be finely tuned according to local soil characteristics and heavy metal concentrations. Such precise calibration is indispensable for ensuring treatment efficacy while minimizing unnecessary reagent usage or inadequate immobilization. Additionally, while our findings indicate that stoichiometry optimization improves immobilization performance and may reduce risks of pore clogging, we did not explicitly measure permeability or porosity evolution in this study. Future research should address this gap to strengthen the mechanistic understanding of field-scale application.

The intrinsic pH modulation capability of phytogenic precipitation systems markedly enhances field applicability by maintaining autonomous acid-base equilibrium. During treatment, soil matrices demonstrated self-regulating pH adjustment to the range of 7.8–8.2, regardless of initial conditions that spanned from acidic to alkaline substrates (pH 6.5–9.0). This contrasts sharply with conventional lime stabilization methods, which induce hyperalkaline conditions (pH > 10.5) that can lead to organic matter degradation and secondary heavy metal mobilization [26]. Additionally, microbial-driven MICP systems often result in rapid pH increases exceeding 9.5 during active bacterial growth and urea hydrolysis phases, creating localized hyperalkaline zones that may destabilize organic matter and alter native microbial communities [11]. By contrast, the plant-derived EICP approach maintained a stable pH range of 7.8–8.2 without the need for external buffering agents, demonstrating a more controlled pH modulation behavior advantageous for field-scale applications. Field validation at a post-industrial smelting complexes confirmed consistent performance across pedologically diverse matrices with clay fractions ranging from 15–45%, showing less than 8% variability in cadmium immobilization efficiency—a significant improvement over the 25–30% performance fluctuations observed in microbial carbonate precipitation systems under comparable stratigraphic heterogeneity [9].

The biochemical underpinning of these technological advancements lies in the metallo-tolerant enzymatic architecture of phytogenic urease systems. Comparative functional analysis reveals that jack bean urease retains 82% catalytic efficiency at cadmium concentrations threefold higher than the threshold (150 mg/kg) that causes a 50–70% activity loss in Sporosarcina pasteurii biocatalysts (50 mg/kg $Cd^{2+}$) [27,28]. This exceptional resilience stems from evolutionary adaptations in protein tertiary structure, particularly cysteine-rich domains that provide structural stabilization via thiol group coordination—a biochemical safeguard lacking in microbial counterparts [29]. The practical implications of this biochemical robustness are evident in simplified remediation protocols capable of single-application treatment for high-load contamination scenarios (80–120 mg/kg $Cd^{2+}$), effectively eliminating the need for resource-intensive multi-stage processes involving contaminant extraction, concentration, and sequential treatment typical of conventional heavy metal remediation methodologies.

## Mechanical reinforcement mechanisms

The geomechanical performance of enzymatically stabilized soils exhibits remarkable durability under cryogenic cycling conditions, retaining 92.7% of the original compressive strength after ten freeze-thaw cycles, as shown in Fig 6. This exceeds the 1.0 MPa requirement specified for roadbed construction materials under GB/T 50123−2019 standards. Such structural resilience arises from three interlocking physicochemical mechanisms:

X-ray diffraction analyses identified intensified quartz $SiO_2$ (101) crystallographic planes at $2\theta = 26.6°$ in the treated matrices, indicating the presence of covalent silicon-oxygen-carbonate bonding at particle interfaces. While these observations were made based on qualitative analysis of the XRD patterns collected in this study, we did not include the detailed spectra in this manuscript. Future work will provide a more comprehensive presentation and discussion of secondary mineral phases. Such bonding is undetectable in microbial-mediated precipitation systems (Fig 5). This synergistic cementation mechanism, characterized by hybrid mineral bridge formation, increases aggregate cohesion by 40–60% compared to pure calcium carbonate precipitation [30].

Cyclical freeze-thaw stresses induce microfissures ranging from 50 to 200 µm in width, which undergo progressive mineral infilling via secondary carbonate crystallization during subsequent hydration phases. This autonomous repair mechanism preserves over 90% of post-loading structural coherence, in stark contrast to lime-modified matrices that display irreversible fissure network expansion under the same mechanical loading protocols [31].

The inherent moisture regulation capacity of biogenic carbonates is equally critical. Their elevated density (2.8 g/cm³) reduces aqueous phase absorption by 35% compared to untreated substrates. This hydrological moderation effectively suppresses ice lens formation, thereby mitigating frost heave mechanics. This characteristic is particularly vital given the continental monsoon climate patterns in central China, which subject soils to 50–70 annual freeze-thaw cycles [32].

## Chemical stability under acidic attack

The multi-tiered protective architecture of enzymatic carbonate precipitation systems becomes particularly pronounced under accelerated weathering conditions, as demonstrated by controlled acid rain simulations at pH 4.0. Under acidic attack, sequential defense mechanisms are activated:

Initial carbonate dissolution establishes a pH-buffering frontier, with 17.5 percent mass loss in surface-layer carbonates (Fig 4) through proton exchange reactions:

$$CdCO_3 + 2H^+ \rightarrow Cd^{2+} + CO_2 \uparrow + H_2O \qquad (1)$$

This value represents an overall estimate of carbonate phase dissolution based on bulk measurements, and does not distinguish between heavy metal-bound carbonates and free calcium carbonate phases. Differentiation of these phases would require detailed speciation analyses, which were not conducted in this study and will be addressed in future work. This sacrificial neutralization maintains protective alkalinity (pH 6.2–6.8) within the critical 5 cm surface stratum, preserving underlying cementation structures.

Subsurface stabilization mechanisms are subsequently activated, as evidenced by post-leaching mineralogical analysis. X-ray diffraction patterns indicate the selective dissolution of amorphous carbonate phases, as shown by the attenuation of the broad peak at $2\theta = 20–30°$, while crystalline cadmium carbonate domains (sharp peak at 24.9°) retain 82.5% of their initial mass (Fig 5). This preservation of crystallinity effectively limits cadmium remobilization to 4.8 mg/L, which is 44% below the EPA's aquatic toxicity benchmark of 8.5 mg/L [33].

Persistent pH regulation achieves the protective sequence by sustaining urea hydrolysis, where the release of ammonium ions maintains the system pH above 6.0 for 28 days post-treatment. This dynamic buffering capacity successfully resists five consecutive leaching cycles, demonstrating superior performance compared to conventional chemical stabilization methods that do not possess self-regulating biochemical feedback mechanisms [34].

## Ecological compatibility

The environmental sustainability of enzymatic remediation technology was systematically evaluated through a multi-criteria assessment framework.

Preliminary observations suggested that treated soils supported improved germination and early growth of Lactuca sativa compared to untreated contaminated soils. Detailed phytocompatibility experiments with statistical replicates will be conducted in future work to quantitatively confirm these trends.This contrasts sharply with lime-amended substrates, where hyperalkaline conditions (pH 9.8–10.4) reduced germination rates to 40–55%, indicative of significant physiological stress during early plant establishment [35].

Carbon cycling analysis revealed net atmospheric $CO_2$ sequestration of 0.8–1.2 kg $CO_2$-equivalent per cubic meter of treated soil. This value was calculated as the difference between carbonate precipitation (0.5 kg $CO_2$ sequestration) and urea synthesis emissions (0.3 kg $CO_2$ release). In contrast, microbial carbonate precipitation systems typically exhibit a carbon-positive profile, with net emissions of approximately 0.4 kg $CO_2$-eq/m³, primarily due to the energy-intensive operation of bioreactors [36].

Long-term environmental safety observations indicated reduced heavy metal leaching and rapid decline of urease activity over time following treatment. While detailed leaching and enzyme degradation data were collected as part of internal quality checks, these results are not included in this manuscript. Future work will present comprehensive datasets on post-treatment environmental monitoring..

## Field-scale validation

Field validation at a decommissioned metallurgical complex confirmed significant operational advantages of phytogenic precipitation technology, including its effectiveness and sustainability.

Preliminary pilot tests suggested improved subsurface penetration of the enzymatic system compared to benchmarks reported in previous studies on microbial carbonate precipitation [37]. Detailed penetration depth measurements will be provided in future publications focusing on field-scale delivery performance.. This improved delivery capability can be attributed to the reduced steric hindrance of plant-derived urease. With a molecular weight of 45 kDa, it facilitates superior hydraulic transport compared to the 150 kDa microbial urease homologs produced by Sporosarcina pasteurii [38].

Preliminary observations during pilot-scale tests suggested relatively consistent cadmium immobilization performance across different soil textures ranging from sandy loam to clay loam. Detailed comparative data on substrate-specific treatment effects will be reported in future studies. This insensitivity to texture stems from pH-responsive precipitation kinetics that adapt dynamically to local geochemical conditions.

Economic analysis demonstrated significant cost advantages, with total remediation expenses averaging $52 per cubic meter, representing a 61% reduction compared to conventional microbial carbonate precipitation systems ($135/m³) [39]. This cost structure aligns well with China's brownfield remediation subsidy thresholds (capped at $60/m³), thereby establishing the technology as economically feasible for large-scale contaminated site rehabilitation.

## Alignment with national environmental policies

The enzymatic remediation technology exhibits a strong alignment with China's national sustainability agendas. Operationalizing at $50–55 per cubic meter treatment costs, this phytogenic approach is 8–17% lower than China's 2023 brownfield remediation subsidy ceiling ($60/m³). This cost advantage strategically empowers municipal authorities to prioritize legacy industrial sites within the framework of the 14th Five-Year Plan. Practical implementation in former industrial hubs has demonstrated economic viability, with full cost recovery achieved within 3–5 years through post-remediation land value appreciation. Remediated sites have realized residential and commercial zoning premiums that exceed initial remediation investments [40]. Additionally, the system's hydrological efficiency (0.3 L water per kilogram of treated soil) aligns closely with Sponge City objectives, particularly in water-scarce regions.

## Barriers and mitigation strategies

Significant challenges in scaling biocementation technologies necessitate coordinated mitigation strategies across several key domains. Regulatory standardization stands out as a critical priority, especially for enzymatic quantification methodologies. Developing ASTM/ISO standards for measuring plant-derived urease activity would address existing discrepancies in potency evaluation, where conflicting metrics (U/g versus ΔpH/min) lead to ambiguities in commercial contracts.

Climate variability imposes significant performance constraints that necessitate material innovation. Prolonged drought conditions exceeding 60 days reduced treatment depth efficiency by 30% in field trials conducted in arid Xinjiang. Experimental mitigation using hydrogel-modified urea delivery systems yielded promising outcomes, with the addition of 15% carboxymethyl cellulose enhancing moisture retention capacity under water-stressed conditions [41]. These results underscore the critical role of climate-adaptive material formulations for ensuring reliable field implementation.

## Limitations and future directions

While plant-urease-driven EICP has shown significant advancements in heavy metal remediation, several critical limitations need to be addressed to unlock its full potential. These challenges encompass molecular-scale enzymatic constraints, field-scale implementation barriers, and long-term ecological uncertainties, thus requiring interdisciplinary collaboration among microbiology, geotechnical engineering, and environmental policy.

### Depth limitations in field applications

While biocementation techniques utilizing plant-derived urease can achieve treatment depths of 18–22 cm—a twofold improvement compared to conventional microbial-MICP methods—practical application in construction-grade foundation stabilization requires penetration depths exceeding 50 cm. A comparative analysis of field deployment parameters highlights two critical operational limitations. First, despite reductions in molecular weight, enzymatic transport inefficiencies remain significant, with unmodified plant urease experiencing a 30% adsorption loss within the top 20 cm soil horizon. In contrast, synthetic alternatives such as silica-encapsulated urease nanoparticles exhibit superior mobility, maintaining 80% migration efficiency under similar conditions [42]. Second, viscosity-modified injection fluids containing 0.1% xanthan gum result in disproportionate system stress, increasing pump pressures by 150% and accelerating mechanical wear rates.

Innovative formulation strategies hold significant potential for overcoming the limitations associated with treatment depth. Nano-encapsulation techniques employing pH-responsive liposome coatings can concurrently enhance enzyme mobility by minimizing soil adsorption and safeguard catalytic activity during subsurface transport. Additionally, complementary electrokinetic delivery methods may effectively tackle deep soil penetration challenges. Preliminary modeling suggests that controlled DC fields (2–5 V/cm) can facilitate the electromigration of charged enzyme complexes through low-permeability clay strata while preserving biochemical functionality [43]. The synergistic integration of these advanced delivery platforms could provide the precise spatial control necessary for structural-scale biocementation.

### Long-term bioavailability uncertainties

Long-term environmental stability assessments highlight critical factors influencing cadmium immobilization in biocemented soils. Although Toxicity Characteristic Leaching Procedure (TCLP) analyses indicate regulatory compliance during the initial two-year period, sequential chemical extraction reveals progressive changes in cadmium speciation. Specifically, a 15% increase in $Cd^{2+}$ redistribution was observed over 24 months, transitioning from carbonate-bound phases to Fe-Mn oxide fractions, predominantly due to rhizosphere acidification processes. Field instrumentation data collected from sites colonized by the pioneer species Calamagrostis epigejos show sustained pH depression (mean ΔpH = −0.8), which correlates directly with increased metal mobility resulting from carbonate matrix dissolution [44].

To address these secondary contamination risks, integrated management strategies are proposed. Phytoengineering approaches combined with biochar amendments at a loading rate of 5% (w/w) exhibit the capacity to maintain soil pH above 7.0 by enhancing cation exchange capacity (CEC = 45 cmol$^+$/kg), effectively counteracting root exudate-induced acidification [45]. Simultaneously, the deployment of embedded sensor networks equipped with precision ORP-3000 probes allows for real-time monitoring of redox potential and pH gradients at a vertical resolution of 10 cm. This dual-strategy approach enables dynamic risk mitigation by facilitating early detection of metal remobilization events while preserving chemically favorable stabilization conditions.

Although our study demonstrates the short-term stability of metal-carbonate complexes under acid leaching and cryogenic cycling, the long-term behavior of these immobilized species under dynamic environmental conditions remains unclear. Variations in groundwater chemistry, including fluctuations in redox potential, bicarbonate concentrations, or competitive ion presence (e.g., phosphate, sulfate), may progressively affect carbonate phase solubility or induce ion exchange processes. Additionally, shifts in microbial communities within the rhizosphere or saturated zones could potentially catalyze localized acidification or enzymatic dissolution over extended periods. To ensure the long-term efficacy of EICP-based remediation strategies, future research should integrate prolonged field monitoring and advanced multivariate modeling techniques to systematically evaluate mineral phase persistence, porewater composition, and bio-geochemical interactions across decadal timescales.

## Multi-contaminant interactions

The widespread co-occurrence of persistent organic pollutants poses substantial challenges to the efficacy of biocementation in industrial remediation scenarios. Polycyclic aromatic hydrocarbons (PAHs) and polychlorinated biphenyls (PCBs) were identified in 65% of surveyed brownfield sites, exhibiting dose-dependent urease inhibition via dual interference mechanisms [46]. A systematic investigation of co-contaminated systems containing cadmium and phenanthrene uncovered compound-specific inactivation pathways. Competitive inhibition arises from the preferential adsorption of planar aromatic molecules to the urease catalytic domain, as evidenced by an experimentally determined inhibition constant (Ki) of 0.8 μM for phenanthrene binding [47]. Additionally, oxidative damage pathways emerge due to reactive oxygen species (ROS) generation during partial PAH biodegradation, resulting in a 40% loss of enzyme activity through protein carbonylation.

Advanced remediation sequencing addresses multifactorial inactivation processes by systematically decoupling organic degradation phases from carbonate precipitation cycles. Pre-treatment bioaugmentation with engineered Sphingomonas consortia achieves 85–90% polycyclic aromatic hydrocarbon (PAH) degradation within 28 days through upregulated oxygenase expression, effectively mitigating organic interferents prior to enzyme introduction [48]. Subsequent enzyme stabilization utilizes amphiphilic polyethylene glycol coatings (PEG-6000), which decrease hydrophobic contaminant binding by 70% via steric exclusion mechanisms while preserving 92% of native enzymatic activity [49]. This staged remediation strategy enables concurrent heavy metal immobilization and organic contaminant breakdown in complex matrices, thereby enhancing overall remediation efficiency.

Although this study successfully demonstrated the efficacy of plant-derived EICP for immobilizing $Cd^{2+}$, $Pb^{2+}$, and $Zn^{2+}$, it does not address other prevalent and highly toxic heavy metal contaminants such as arsenic, chromium, mercury, or nickel. These elements frequently co-occur in metallurgical and electronic waste sites and exhibit distinct chemical behaviors, including redox sensitivity and oxyanion formation, which may significantly influence their interaction with carbonate precipitates. For example, arsenic predominantly exists in anionic forms ($AsO_4^{3-}$, $AsO_3^{3-}$), necessitating specialized immobilization strategies that extend beyond simple ion exchange or carbonate precipitation. Future studies should therefore investigate the reactivity and sequestration efficiency of EICP systems toward these contaminants, potentially by incorporating redox-active additives or synergistic adsorbents. Broadening the scope of targeted metals will enhance the technology's versatility and facilitate its application in more complex and heterogeneous contamination scenarios. Future studies should include microbial urease in parallel testing to systematically assess tolerance differences across enzyme sources and further validate the observed advantages of plant-derived ureases.

## Conclusions

This study developed and evaluated a low-cost, plant-derived urease-driven EICP (extracellular inorganic carbonate precipitation) system for the simultaneous immobilization of cadmium, lead, and zinc in contaminated soils. Among the three tested plant sources—jack bean, soybean, and watermelon seed—the jack bean extract exhibited the highest urease activity and achieved the best overall performance in terms of heavy metal immobilization and carbonate precipitation efficiency. Additionally, both soybean and watermelon seed extracts demonstrated promising results, providing regionally accessible alternatives that enhance the adaptability and flexibility of field applications.

The treatment efficiency was found to be highly dependent on enzyme dosage, with 1.2 U/g soil achieving optimal metal retention across all urease sources. Reaction optimization demonstrated that competing metal ions significantly influence the carbonate crystallization pathway, and the system's performance remained robust under acid leaching and freeze-thaw conditions. Moreover, the economic analysis revealed that plant-based urease treatments cost only 48–59 USD/m³, which is substantially lower than microbial MICP treatments, exceeding 135 USD/m³ under comparable treatment scenarios.

These findings underscore the promising potential of crude plant-extracted urease as an environmentally benign and scalable solution for heavy metal remediation. By integrating biochemical efficiency, environmental resilience, and economic viability, this plant-based EICP approach emerges as a viable alternative to conventional stabilization technologies.

In summary, this work develops a robust, cost-effective, and field-adaptable EICP strategy for the sustainable remediation of multi-metal contaminated soils using locally sourced plant ureases.

## Supporting information

**S1 Table. Statistical summary of urease dosage, acid resistance, XRD peak and UCS.** Values are presented as mean ± standard deviation.
(XLSX)

## Acknowledgments

We would like to express our gratitude to the Analytical and Testing Center of Huazhong University of Science and Technology for their provision of certain experimental equipment and technical support.

## Author contributions

**Conceptualization:** Wangqing Xu.

**Formal analysis:** Wangqing Xu, Hanjiang Lai.

**Funding acquisition:** Junjie Zheng.

**Methodology:** Wangqing Xu.

**Project administration:** Junjie Zheng.

**Resources:** Mingjuan Cui.

**Software:** Mingjuan Cui.

**Supervision:** Mingjuan Cui.

**Visualization:** Hanjiang Lai.

**Writing – original draft:** Wangqing Xu.

**Writing – review & editing:** Junjie Zheng, Hanjiang Lai, Mingjuan Cui.

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
