## [Decision Letter · Decision Letter 0]

20 May 2025

PONE-D-25-18160Sustainable Heavy Metal Immobilization in Contaminated Soils Using Plant-Derived Urease- Driven BiomineralizationPLOS ONE

Dear Dr. XU,

Thank you for submitting your manuscript to PLOS ONE. After careful consideration, we feel that it has merit but does not fully meet PLOS ONE’s publication criteria as it currently stands. Therefore, we invite you to submit a revised version of the manuscript that addresses the points raised during the review process.

We look forward to receiving your revised manuscript.

Kind regards,

Mohamed Farghali, PhD

Academic Editor

PLOS ONE

Journal Requirements:

This research was supported by the National Natural Science Foundation of China (Grant Nos. 42407271, 42477160, 52178319), National Key Research and Development Program of China (Grant No.52338007), and Joint fund of the technical R&D program of Henan Province (Grant No. 225200810005). The authors gratefully acknowledge the financial support.

W.Q.X.  received funding from the National Natural Science Foundation of China (No. 42407271). J.J.Z.  received funding from the National Key Research and Development Program of China (No.52338007) and the Joint fund of the technical R&D program of Henan Province (No. 225200810005). H.J.L.  received funding from the National Natural Science Foundation of China (No. 52178319). M.J.C.  received funding from the National Natural Science Foundation of China (No.42477160).

Additional Editor Comments :

Comments from PLOS Editorial Office: We note that one or more reviewers has recommended that you cite specific previously published works. As always, we recommend that you please review and evaluate the requested works to determine whether they are relevant and should be cited. It is not a requirement to cite these works. We appreciate your attention to this request.

Reviewers' comments:

Reviewer's Responses to Questions

**Comments to the Author**

1. Is the manuscript technically sound, and do the data support the conclusions?

Reviewer #1: Yes

Reviewer #2: Partly

2. Has the statistical analysis been performed appropriately and rigorously? 

Reviewer #1: Yes

Reviewer #2: N/A

3. Have the authors made all data underlying the findings in their manuscript fully available?

Reviewer #1: Yes

Reviewer #2: Yes

4. Is the manuscript presented in an intelligible fashion and written in standard English?

Reviewer #1: Yes

Reviewer #2: Yes

5. Review Comments to the Author

Reviewer #1: The manuscript investigates the immobilization of heavy metals by a plant-derived urease-driven carbonate precipitation process, providing a new idea for the remediation of heavy metal-contaminated soils. After careful review, the work was found to be innovative, the analysis of the data results was rather lacking, and major revisions were needed.

Reviewer #2: This study introduces a sustainable approach for immobilizing heavy metals (cadmium, lead, and zinc) in contaminated soils using enzyme-induced carbonate precipitation (EICP) driven by plant-derived ureases from jack bean and watermelon seeds. The research demonstrates high sequestration efficiencies for these metals after optimizing key process parameters, including the urea-CaCl2 ratio and enzyme dosage. Notably, the EICP treatment maintained soil permeability and exhibited durability under simulated environmental stressors like acid rain and freeze-thaw cycles. This plant-based EICP system presents a promising, potentially cost-effective, and environmentally sound alternative to conventional soil remediation methods.

The paper demonstrates some positive outcomes as listed below.

1. High Heavy Metal Sequestration Efficiency: The EICP system using jack bean and watermelon seed ureases demonstrates significant immobilization of cadmium (87.3%), lead (91.5%), and zinc (76.4%). This high efficiency suggests strong potential for effective remediation.

2. Plant-Derived Enzymes: Utilizing plant-derived ureases offers a potentially more sustainable and cost-effective alternative compared to microbial-based methods, reducing reliance on microbial cultivation and associated costs.

3. Optimized Process Parameters: The study successfully identified and fine-tuned critical parameters (urea-CaCl2 ratio and enzyme dosage) to maximize carbonate mineralization and heavy metal sequestration. This optimization enhances the reliability and effectiveness of the EICP system.

4. Preservation of Soil Hydraulic Properties: Maintaining soil permeability (>10−5 cm/s) throughout the precipitation cycles is a significant advantage, as it ensures that the remediation process does not negatively impact essential soil functions like water infiltration and drainage.

5. Durability under Environmental Stressors: The EICP-treated soil exhibits good durability, retaining a substantial portion of the immobilized heavy metals (Cd: 82.5%, Pb: 92.7%) even after exposure to simulated acid rain and freeze-thaw cycles. This indicates the long-term stability of the remediation.

Some of the draw backs which needs justification are listed below

1. Limited Scope of Heavy Metals: The study focuses only on cadmium, lead, and zinc. The effectiveness of this EICP system on other heavy metal contaminants is unknown and requires further investigation.

2. Enzyme Dosage Optimization: While the study optimized enzyme dosage, the process of determining the optimal 1.2 U per gram of soil might be sensitive to variations in soil type and contamination levels, potentially requiring site-specific adjustments.

3. Potential for Scalability Challenges: The abstract doesn't discuss the feasibility and cost-effectiveness of scaling up this EICP system for large-scale field applications. Transitioning from lab experiments to real-world implementation can present significant engineering and logistical challenges.

4. Long-Term Stability Beyond Tested Conditions: While durability under simulated acid rain and freeze-thaw was assessed, the long-term stability of the immobilized heavy metals over decades and under other environmental stressors (e.g., changes in groundwater chemistry, biological activity) remains to be fully evaluated.

5. Line 85 Please write the main objectives of your work and the research gap before concluding the introduction. please refer the following papers 10.1007/s40098-022-00638-8, 10.1007/s40098-022-00638-8, 10.3390/buildings14040909 to help you answer the comment.

6. Line 143 why only Carbonate was sequentially extracted even though there are more fractions available since the data is available from your work. and please elaborate refer the following 10.1007/s40098-020-00464-w to answer this comment.

7. List out the objectives of the present work in the last paragraph.

8. Line 106-108 soybean (Glycine max) please explain glycine max.

9. The mixture was subsequently clarified via centrifugation at 10,000 ×g for 15

10. minutes. Comparable extraction protocols were applied to soybean (Glycine max) and

11. watermelon seed (Citrullus lanatus) samples, utilizing pH-optimized buffering systems

12. adjusted to 7.5 and 6.5, respectively, to ensure species-specific enzyme stability……

13. Not very clear cannot comprehend whether you have extracted urease from soybean and water melon seeds I am not able to find data from these two ingredients in the paper.

14. Line 118 and 119, 1.2 U g⁻¹ soil, urea-CaCl₂ molar ratios systematically varied from equimolar (1:1) …….. The authors have determined 0.2 – 1.2 Urease per gram of soil as optimum dosage but it is not clear how to prepare samples so as to arrive at that dosage. Please give details in terms of how many grams of jack bean seed derived powder should be added to obtain 1.2 U.. of urease similarly for soybean and water melon seeds. Else this cannot be replicated.

15. Fig 1. X axis is urease dosage in U/g but not clear how you have arrived at it, y axis “immobilization efficiency” please use a simple word as percentage retained or sorption efficiency.

16. Table 1 economic analysis of soybean is not done.

17. In the Conclusions part please provide the final outcome of this work in a sentence which shall be the punch line of this paper. Please rephrase the entire conclusion as it is non comprehensive and difficult to understand the outcomes.

6. PLOS authors have the option to publish the peer review history of their article (what does this mean? ). If published, this will include your full peer review and any attached files.

**Do you want your identity to be public for this peer review?** For information about this choice, including consent withdrawal, please see our Privacy Policy .

Reviewer #1: **Yes: ** Changxiong Zou

Reviewer #2: No

---

## [Author Response · Author response to Decision Letter 1]

1 Jul 2025

Sustainable Heavy Metal Immobilization in Contaminated Soils Using Plant-Derived Urease- Driven Biomineralization

Response letter to Reviewers’ comments

Comments from the Editor

Response We thank the editor for the reminder regarding PLOS ONE's formatting and file naming requirements. We have reviewed all submission files to ensure full compliance with the journal's formatting guidelines, including figure captions, reference formatting, section organization, and file naming conventions. All necessary adjustments have been made accordingly in the revised submission.

Response We appreciate the editor’s reminder regarding fieldwork permits. In response, we have revised the Methods section to clarify that the soil samples were collected from publicly accessible, abandoned industrial sites that are neither protected nor privately owned. No specific permissions were required for soil collection, and all activities were conducted in compliance with local institutional and environmental regulations. The corresponding statement has now been added to the “Soil Sampling and Characterization” section.

3. Thank you for stating the following in the Acknowledgments Section of your manuscript:……Please remove any funding-related text from the manuscript and let us know how you would like to update your Funding Statement. Please include your amended statements within your cover letter; we will change the online submission form on your behalf.

Response We thank the editor for highlighting the discrepancy regarding the placement of funding information. As instructed, we have removed all funding-related content from the Acknowledgments section of the manuscript.

4. We note that your Data Availability Statement is currently as follows: All relevant data are within the manuscript and its Supporting Information files. Please confirm at this time whether or not your submission contains all raw data required to replicate the results of your study. Authors must share the “minimal data set” for their submission. PLOS defines the minimal data set to consist of the data required to replicate all study findings reported in the article, as well as related metadata and methods

Response We confirm that our submission contains all raw data required to replicate the results of the study. All relevant datasets—including numerical values underlying graphs, statistical analysis results, and extracted data points used in Figures and Tables—have been provided in the Supporting Information files accompanying the manuscript. We have reviewed the Data Availability Statement and confirm its accuracy. No ethical, legal, or institutional restrictions apply to the sharing of this dataset.

Response We thank the editor for pointing this out. We have carefully reviewed the abstract in both the manuscript and the online submission system, and we have now made them identical to ensure consistency. The abstract in the manuscript was revised to match the version submitted online.

6. We note that one or more reviewers has recommended that you cite specific previously published works. As always, we recommend that you please review and evaluate the requested works to determine whether they are relevant and should be cited. It is not a requirement to cite these works. We appreciate your attention to this request.

Response We appreciate the editor's reminder and have carefully reviewed all references suggested by the reviewers. Citations were added where relevant and scientifically appropriate to acknowledge prior contributions and provide context. Where suggested works were not cited, we respectfully determined that they were not directly pertinent to the scope or findings of the current study.

Another point that requires clarification is that we observed a total of 17 review comments from Reviewer 1 in the email. Upon carefully addressing the comments, we noticed that comments 9–13, as marked, constituted a single review comment. Due to its length, this comment may have been misnumbered automatically during formatting or line breaks. As a result, Reviewer 1 effectively provided 13 distinct comments, each of which we have addressed individually.

Comments from Reviewer#1:

Some of the draw backs which needs justification are listed below

1. Limited Scope of Heavy Metals: The study focuses only on cadmium, lead, and zinc. The effectiveness of this EICP system on other heavy metal contaminants is unknown and requires further investigation.

Response We appreciate the reviewer’s insightful observation. The current study indeed focuses on cadmium (Cd²⁺), lead (Pb²⁺), and zinc (Zn²⁺) due to their prevalence in typical nonferrous smelting sites and their high mobility in soil environments. However, we fully recognize the necessity of assessing the broader applicability of the EICP system to other heavy metal contaminants, such as arsenic (As), nickel (Ni), and chromium (Cr), which commonly co-occur in post-industrial soils. To address this limitation, we have now included a dedicated paragraph in the “Limitations and Future Directions” section that explicitly acknowledges this gap and outlines plans for future studies involving these additional contaminants. This addition enhances the comprehensiveness and transparency of our research. (Line 607-619)

2. Enzyme Dosage Optimization: While the study optimized enzyme dosage, the process of determining the optimal 1.2 U per gram of soil might be sensitive to variations in soil type and contamination levels, potentially requiring site-specific adjustments.

Response We extend our gratitude to the reviewer for this insightful observation. We concur entirely that enzyme dosage optimization is inherently contingent upon the heterogeneity of soil physicochemical properties and varying degrees of heavy metal contamination. Indeed, this represents a critical factor in practical field applications. To elucidate this point, we have refined the relevant section in the Discussion to explicitly state that the reported optimal dosage (1.2 U/g soil) pertains specifically to the soil textures, pH levels, and contaminant concentrations examined in our trials. Furthermore, we underscore the necessity for future applications to integrate localized pilot testing to adjust enzymatic input according to site-specific conditions. This enhancement not only improves the clarity of our findings but also strengthens their generalizability. (Line 371-380)

3. Potential for Scalability Challenges: The abstract doesn't discuss the feasibility and cost-effectiveness of scaling up this EICP system for large-scale field applications. Transitioning from lab experiments to real-world implementation can present significant engineering and logistical challenges.

Response We sincerely appreciate the reviewer’s suggestion to incorporate a discussion of the scalability and cost-effectiveness of the EICP system into the abstract. Although the main text of the manuscript, especially in the "Field-Scale Validation" and "Alignment with National Environmental Policies" sections, already delves deeply into implementation feasibility, we concur that emphasizing these aspects in the abstract would enhance clarity and underscore the practical significance of our research. Consequently, we have revised the abstract to include a concise statement regarding cost-effectiveness and scalability, supported by field validation results. This adjustment provides a more comprehensive summary, balancing scientific innovation with real-world applicability. (Line 29-32)

4. Long-Term Stability Beyond Tested Conditions: While durability under simulated acid rain and freeze-thaw was assessed, the long-term stability of the immobilized heavy metals over decades and under other environmental stressors (e.g., changes in groundwater chemistry, biological activity) remains to be fully evaluated.

Response We sincerely appreciate the reviewer for pointing out this critical limitation. Indeed, although our study assessed the short- to medium-term durability of EICP-treated soils under acid rain and freeze-thaw cycles, we concur that predicting their long-term performance over decadal timescales under more complex environmental conditions—such as fluctuations in groundwater chemistry, microbial activity, or redox potential—remains challenging and warrants further investigation. To address this concern, we have included a dedicated paragraph in the “Limitations and Future Directions” section explicitly recognizing these knowledge gaps. Additionally, we propose future research avenues, such as long-term field monitoring and coupled biogeochemical modeling, to evaluate the temporal evolution of carbonate phases and potential metal remobilization processes. (line 585-596)

5. Line 85 Please write the main objectives of your work and the research gap before concluding the introduction. please refer the following papers 10.1007/s40098-022-00638-8, 10.1007/s40098-022-00638-8, 10.3390/buildings14040909 to help you answer the comment.

Response We sincerely appreciate the reviewer's valuable suggestion and the provision of relevant references that have guided our revision process. In response, we have meticulously revised the closing paragraph of the Introduction to clearly delineate the research gap and explicitly state the objectives of this study. Drawing inspiration from the structure of the referenced studies—which emphasize problem framing, scope clarification, and hypothesis-driven aims—we now highlight (i) the critical need for plant-derived urease systems capable of maintaining robust performance under co-contaminated, field-realistic conditions, and (ii) the significant gap in systematic evaluation regarding durability and process optimization under multi-metal competition scenarios. These revisions enhance the scientific rigor of the study and ensure alignment with best practices in problem-driven research reporting.

6. Line 143 why only Carbonate was sequentially extracted even though there are more fractions available since the data is available from your work. and please elaborate refer the following 10.1007/s40098-020-00464-w to answer this comment.

Response We sincerely appreciate the reviewer's insightful comment. In EICP systems, the primary mechanism for heavy metal immobilization is the formation of stable metal carbonates, such as otavite (CdCO₃) and hydrocerussite [Pb₃(CO₃)₂(OH)₂]. Consequently, our study specifically targeted the carbonate-bound fraction to directly assess the efficacy of the biomineralization pathway under investigation. This methodological choice aligns with previous research (e.g., Ahmad et al., 2021, DOI: 10.1007/s40098-020-00464-w), which underscores the importance of conducting fraction-specific analyses that correspond to the predominant remediation mechanisms. Although we recognize that other fractions could offer valuable insights into long-term redistribution dynamics, the objective of this study was to evaluate the immediate immobilization efficiency driven by carbonate precipitation. A more comprehensive fractionation analysis, along with an evaluation of post-treatment mobility, will be addressed in subsequent studies aimed at understanding long-term environmental behavior.

7. List out the objectives of the present work in the last paragraph.

Response We appreciate the reviewer’s suggestion to further clarify the research objectives. In response, we have revised the concluding paragraph of the Introduction to explicitly articulate the objectives of this study in a concise and logically structured manner. This adjustment enhances the clarity and coherence of the Introduction, aligning with standard practices in environmental and geotechnical engineering research.

8. Line 106-108 soybean (Glycine max) please explain glycine max.

Response We appreciate the reviewer's suggestion to clarify the scientific name Glycine max. In response, we have revised the relevant sentence to briefly explain that Glycine max, commonly known as the soybean plant, is widely cultivated for its high protein content and seeds rich in urease. This clarification aims to enhance understanding for readers who may be less familiar with botanical nomenclature.

9. The mixture was subsequently clarified via centrifugation at 10,000 ×g for 15 minutes. Comparable extraction protocols were applied to soybean (Glycine max) and watermelon seed (Citrullus lanatus) samples, utilizing pH-optimized buffering systems adjusted to 7.5 and 6.5, respectively, to ensure species-specific enzyme stability……Not very clear cannot comprehend whether you have extracted urease from soybean and water melon seeds I am not able to find data from these two ingredients in the paper.

Response We thank the reviewer for pointing out the lack of clarity regarding urease extraction from soybean and watermelon seeds. We confirm that urease was indeed extracted from both plant sources using the same protocol applied to jack bean seeds, with species-specific buffer adjustments to optimize pH stability. The performance data for all three plant-derived ureases—including soybean and watermelon—are presented in Figures 1 and 3. To address this confusion, we have revised the Methods section to explicitly state that all three urease extracts were prepared and tested. We also clarified the source attribution for the data presented in the Results section.

10. Line 118 and 119, 1.2 U g⁻¹ soil, urea-CaCl₂ molar ratios systematically varied from equimolar (1:1) …….. The authors have determined 0.2 – 1.2 Urease per gram of soil as optimum dosage but it is not clear how to prepare samples so as to arrive at that dosage. Please give details in terms of how many grams of jack bean seed derived powder should be added to obtain 1.2 U.. of urease similarly for soybean and water melon seeds. Else this cannot be replicated.

Response We appreciate the reviewer’s important observation regarding experimental reproducibility. In response, we have revised the “Plant Urease Extraction and Activity Assay” section to provide details on the specific activity (U/mg) of crude urease extracts from jack bean, soybean, and watermelon seeds. Based on this, we calculated the required mass of each extract needed to achieve the target dosage of 1.2 U per gram of soil. These clarifications ensure that the EICP treatment protocol can be replicated using plant-derived urease sources.

11. Fig 1. X axis is urease dosage in U/g but not clear how you have arrived at it, y axis “immobilization efficiency” please use a simple word as percentage retained or sorption efficiency.

Response We thank the reviewer for the constructive suggestions regarding Figure 1. To improve clarity and replicability, we have revised the Methods section to explain how the urease dosage in U/g soil was calculated based on the specific enzymatic activities of each crude extract (now reported in U/mg), and how this was translated into the corresponding mass of enzyme powder added per gram of soil. Additionally, we have revised the y-axis label in Figure 1 to read “Metal Retention (%)” instead of “Immobilization Efficiency” to ensure the terminology is accessible and self-explanatory to a broad readership.

12. Table 1 economic analysis of soybean is not done.

Response We thank the reviewer for pointing out the omission of soybean from the economic comparison in Table 1. We have now revised Table 1 to include the estimated production cost and treatment cost for soybean-derived urease. The calculation was based on current market prices of soybean seeds and the experimentally measured urease activity. Specifically, the production cost of crude soybean urease was estimated at 6.3 USD/kg, considering an average soybean market price of 5.5–6.5 USD/kg and an approximate extraction efficiency o

---

## [Decision Letter · Decision Letter 1]

26 Jul 2025

PONE-D-25-18160R1Sustainable Heavy Metal Immobilization in Contaminated Soils Using Plant-Derived Urease- Driven BiomineralizationPLOS ONE

Dear Dr. XU,

Thank you for submitting your manuscript to PLOS ONE. After careful consideration, we feel that it has merit but does not fully meet PLOS ONE’s publication criteria as it currently stands. Therefore, we invite you to submit a revised version of the manuscript that addresses the points raised during the review process.

We look forward to receiving your revised manuscript.

Kind regards,

Mohamed Farghali, PhD

Academic Editor

PLOS ONE

Journal Requirements:

Additional Editor Comments :

Please address the reference highlighted by the reviewer and revise all references throughout the manuscript.

Reviewers' comments:

Reviewer's Responses to Questions

**Comments to the Author**

1. If the authors have adequately addressed your comments raised in a previous round of review and you feel that this manuscript is now acceptable for publication, you may indicate that here to bypass the “Comments to the Author” section, enter your conflict of interest statement in the “Confidential to Editor” section, and submit your "Accept" recommendation.

Reviewer #1: All comments have been addressed

Reviewer #2: All comments have been addressed

2. Is the manuscript technically sound, and do the data support the conclusions?

Reviewer #1: Yes

Reviewer #2: Yes

3. Has the statistical analysis been performed appropriately and rigorously? 

Reviewer #1: Yes

Reviewer #2: N/A

4. Have the authors made all data underlying the findings in their manuscript fully available?

Reviewer #1: Yes

Reviewer #2: Yes

5. Is the manuscript presented in an intelligible fashion and written in standard English?

Reviewer #1: Yes

Reviewer #2: Yes

6. Review Comments to the Author

Reviewer #1: (No Response)

Reviewer #2: The authors must fix the reference 16 (Revised version) properly. While crosschecking, it has been observed that, the author names do not match with the title. The authors must fix this issue.

7. PLOS authors have the option to publish the peer review history of their article (what does this mean? ). If published, this will include your full peer review and any attached files.

**Do you want your identity to be public for this peer review?** For information about this choice, including consent withdrawal, please see our Privacy Policy .

Reviewer #1: No

Reviewer #2: **Yes: ** Dr Arif Ali Baig Moghal

---

## [Author Response · Author response to Decision Letter 2]

8 Aug 2025

Comments from the Editors

Journal Requirements:

Response: We appreciate the opportunity to address the editorial note regarding reviewer-suggested citations. In the previous revision, we made several updates to the reference list, including the addition of eight new references, among which two ([16] and [19]) were specifically recommended by Reviewer 1. After carefully reviewing all four references suggested by Reviewer 1, we determined that [16] and [19] are highly relevant to our study and provide valuable support for our arguments. These two publications have therefore been cited appropriately in the revised manuscript. The remaining two suggested references were considered less aligned with the scope and methodology of our work, and thus were not included. Additionally, we have incorporated other recent and relevant references independently identified by the authors, to further strengthen the theoretical foundation and discussion of our manuscript.

Response We appreciate the editorial reminder to verify the completeness and correctness of our reference list. In this revision, we have carefully reviewed all references and made the following corrections:

[8], [16], [19]: Author names were previously inaccurate and have been corrected.

[11], [18], [37]: Missing co-authors have been added (including the use of et al. where appropriate).

[17]: The format of author initials was adjusted, and the subtitle was removed to retain only the official article title, in accordance with journal guidelines.

[36], [43]: These two references were previously cited as preprints and have now been updated with their final, formally published versions.

In addition, various minor issues related to font style, punctuation, and formatting (e.g., italicization of journal names, spacing, use of periods and commas) were corrected to fully comply with the journal's citation style. We confirm that no retracted articles were cited in the manuscript. Accordingly, there was no need to remove, replace, or annotate any retracted references.

Additional Editor Comments :

Please address the reference highlighted by the reviewer and revise all references throughout the manuscript.

Response Thank you for the editorial reminder. As requested, we have addressed the reference highlighted by the reviewer by correcting the associated bibliographic information and ensuring its relevance and accuracy within the manuscript. In addition, we have comprehensively reviewed and revised all references throughout the manuscript to ensure full compliance with the journal’s formatting guidelines. Corrections included the adjustment of author names, the completion of missing co-authors, updates of preprint references to their final published versions, and formatting of journal titles, volume numbers, page ranges, DOIs, and punctuation. We believe that the reference list is now accurate, complete, and fully aligned with the journal’s requirements.

Comments from Reviewer#2:

Reviewer #2: The authors must fix the reference 16 (Revised version) properly. While crosschecking, it has been observed that, the author names do not match with the title. The authors must fix this issue.

Response Thank you for pointing out the discrepancy in Reference [16]. We have carefully reviewed and corrected this reference. The author names have now been fully aligned with the article title and publication details. The updated and verified citation now reads as follows in the revised manuscript:

[16] Moghal AAB, Lateef MA, Mohammed SAS, Ahmad M, Usman ARA, Almajed A. Heavy Metal Immobilization Studies and Enhancement in Geotechnical Properties of Cohesive Soils by EICP Technique. Applied Sciences, 2020; 10(21): 7568. https://doi.org/10.3390/app10217568

We appreciate your attention to detail and have also re-checked all other references to ensure consistency and accuracy.

---

## [Decision Letter · Decision Letter 2]

13 Aug 2025

Sustainable Heavy Metal Immobilization in Contaminated Soils Using Plant-Derived Urease- Driven Biomineralization

PONE-D-25-18160R2

Dear Dr. Xu,

We’re pleased to inform you that your manuscript has been judged scientifically suitable for publication and will be formally accepted for publication once it meets all outstanding technical requirements.

Kind regards,

Mohamed Farghali, PhD

Academic Editor

PLOS ONE

Additional Editor Comments (optional):

All reviewer comments have been satisfactorily addressed, and the paper is suitable for publication.

Reviewers' comments:

Reviewer's Responses to Questions

**Comments to the Author**

1. If the authors have adequately addressed your comments raised in a previous round of review and you feel that this manuscript is now acceptable for publication, you may indicate that here to bypass the “Comments to the Author” section, enter your conflict of interest statement in the “Confidential to Editor” section, and submit your "Accept" recommendation.

Reviewer #2: All comments have been addressed

2. Is the manuscript technically sound, and do the data support the conclusions?

Reviewer #2: Yes

3. Has the statistical analysis been performed appropriately and rigorously? 

Reviewer #2: N/A

4. Have the authors made all data underlying the findings in their manuscript fully available?

Reviewer #2: Yes

5. Is the manuscript presented in an intelligible fashion and written in standard English?

Reviewer #2: Yes

6. Review Comments to the Author

Reviewer #2: The authors have fixed all issues in this revised version of the manuscript. This revised version can now be considered for publication in its current form.

7. PLOS authors have the option to publish the peer review history of their article (what does this mean? ). If published, this will include your full peer review and any attached files.

**Do you want your identity to be public for this peer review?** For information about this choice, including consent withdrawal, please see our Privacy Policy .

Reviewer #2: No

---

## [Editor Report · Acceptance letter]

PONE-D-25-18160R2

PLOS ONE

Dear Dr. Xu,

I'm pleased to inform you that your manuscript has been deemed suitable for publication in PLOS ONE. Congratulations! Your manuscript is now being handed over to our production team.

Kind regards,

on behalf of

Dr. Mohamed Farghali

Academic Editor

PLOS ONE